# Sounds, Emotions, and the Body in Pentecostal Romani Communities in Slovakia

**Jana Belišová** 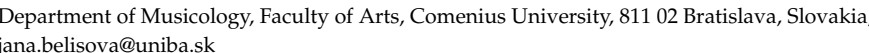

Department of Musicology, Faculty of Arts, Comenius University, 811 02 Bratislava, Slovakia;
jana.belisova@uniba.sk

**Abstract:** In the past, the Romani in Slovakia identified with the prevailing religion, mainly with the Roman Catholic Church. However, the missionary activities of various Christian denominations after 1990 resulted in the conversion of the Romani to Pentecostal Christian communities. This launched a long, creative process of the formation of Pentecostal Romani music. Romani believers consider music and the ability to play and sing to be a gift from God and view these as a form of prayer that should serve for the praise of God. That is why many have given up their worldly music making and now play only praise songs. They gradually modified the hymns they borrowed and replaced them with their own creations. The soundscape of religion does not lie only in religious singing and music, as the emotional sermons and prayers, glossolalia and sounds during the healing and blessing rituals can also be considered religious sounds. During the worship services, this mixture of various sounds leads to the gradual spiritual and emotional unification of the community. The music and the rituals create feelings of intense sensory and emotional character that reflect in bodily expressions. Movements, dance, and the positions of the hands can help glorify God and experience the worship service more intensely. However, under certain circumstances, they might become sources of temptation and sin. This is related to the concepts of "purity" and "impurity". The premises, whether sacral or profane, interior or exterior ones, also play a significant role in creating the sound. In writing this paper, I have also drawn on my own research on Romani Christian songs, which I carried out in (2012–2013 in Eastern Slovakia).

**Keywords:** Pentecostal services; praise songs; worship songs; gospel; sound; emotions; prayer; body



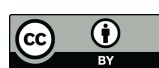

Then Ilúvatar said to them: Of the theme that I have declared to you, I will now that ye make in harmony together a Great Music. And since I have kindled you with the Flame Imperishable, ye shall show forth your powers in adorning this theme, each with his own thoughts and devices, if he will. But I will sit and hearken, and be glad that through you great beauty has been wakened into song.

Then the voices of the Ainur, like unto harps and lutes, and pipes and trumpets, and viols and organs, and like unto countless choirs singing with words, began to fashion the theme of Ilúvatar to a great music; and a sound arose of endless interchanging melodies woven in harmony that passed beyond hearing into the depths and into the heights, and the places of the dwelling of Ilúvatar were filled to overflowing, and the music and the echo of the music went out into the Void, and it was not void.[1]

Tolkien, J.R.R.

## 1. Introduction

In the past three decades, the topic of the Romani and religion has aroused interest and has been quite often discussed in Slovakia. This must be partly because great changes and a revival can be observed in the spiritual and religious lives of the Romani, and these are proving to be lasting ones, with a noticeably positive impact on their lives in several

areas. Many Romani have converted to Pentecostal churches and communities or are introducing Pentecostal elements into the liturgical services of traditional churches. In my paper, I would like to consider why the Romani prefer the Pentecostal character to the traditional way of worship and what benefits it is bringing to their spiritual lives. By joining Pentecostal communities, their occasional, passive church attendance has changed to sincere involvement. They are discovering that music, which is very important to them, can become a form of prayer, and they can pray by singing even at home. The opportunity to introduce elements of their own culture into the liturgy, including their language and music, is also important. Conversion also has an impact on their everyday lives—by giving up their previous way of life, many problems are solved, and their lives improve socially and economically. They are accepted in the new communities, and they are given respect and responsibility.

In writing this paper, I have mostly relied on my own research of many years among Romani communities in Slovakia and especially on my research on Romani Christian songs, which I carried out in 2012–2013 in several locations in Eastern and Middle Slovakia. The outcomes of this research appeared in two publications. The first one contains two studies by two authors, mine approaching the topic primarily from the ethnomusicological perspective, zooming in on the liturgical and non-liturgical sacred music of the Romani, the role of the pastor, and the origin and characteristics of these songs. By the method of oral history, the second study processes interviews with believers and priests or pastors about the beginnings of their pastoral ministry among the Romani, about individual congregations, and about the personal experiences of the believers (Belišová and Mojžišová 2014). The core of the second output is a collection of Romani Christian songs (Belišová 2015). I supplemented and clarified my thoughts and observations, which are based on the processing of the interviews and the songs from the research, by adding testimonies of respondents—believers and the clergy. The research consisted not only of gathering materials, but also of attending worship services of various types and occasionally even participating in the everyday lives of the respondents, and of participant and non-participant observation.

In processing this topic, I relied on the publications and pieces of research that have been published on this topic in Slovakia. Partial information about their faith can be found in several monographs about the Romani, and some authors have sporadically published articles about them in periodicals and conference proceedings. One of the first of these was Eva Davidová's article on the folk religion of the Romani near Trebišov (Davidová 1998), and Zuzana Palubová (2001) addressed the folk religion of the Romani in Levoča. Emília Horváthová (1964) and, later, Arne B. Mann (2002) discussed the religious ideas and acts of the spiritual life of the Romani. The first comprehensive volume, which presents a picture of the spirituality of the Slovak Romani through the contributions of several authors, is the publication *Boh všetko vidí [God Sees Everything]* (Kováč and Mann 2003). In addition to ethnographic and religious topics, such as the magical protection of the newborn, the phenomenon of death in folk religion, the sanctity of oaths, and so on, the book also contains a contribution by Jakub Mináč, *Rómske nové duchovné piesne v obci Rankovce [New Romani Hymns in the Village of Rankovce]*, which is already close to my topic. The activities of the churches and religious movements among the Romani in Slovakia were first mapped comprehensively by the "Social Inclusion of the Romani through Religion" project, whose results confirm some of the above observations published in the book *Boh medzi bariérami [God between Barriers]* (Podolinská and Hrustič 2010). Its editors had dealt with the topic of Romani religion for many years, and they published the results of their research in several publications.

The Romani Pentecostalism: Gypsies and Charismatic Christianity anthology of 2014 presents for the first time the papers of fifteen scholars who studied local variants of Romani Pentecostalism in various parts of Europe: Acton and Laurent (UK), Marushiakova and Popov (former Soviet Union), Slavkova (Bulgaria), Delgado (Spain), Podolinská and Hrustič (Slovakia), Åberg and Thurfjell (Finland), and Kwiek (Lovari and Kelderaši). In sixteen chapters, it elucidates the multifaceted and complex phenomenon of today's Romani

Pentecostalism (*Romani Pentecostalism: Gypsies and Charismatic Christianity* 2014). Most authors of the papers in this anthology had been dealing with this topic for many years (Slavkova 2012; Marushiakova and Popov 2014; Åberg 2014).

More literature on this subject and a better processing of the topic of Romani religious music in Pentecostal communities can be found abroad. The Czech Republic is the closest geographically and culturally, and many Romani from Slovakia have moved there, while they maintain close contact with their Slovak relatives and thus have many ties in both their cultural and spiritual life. The Romani Pentecostal movement and Romani Christian songs in the Czech Republic were addressed by the ethnomusicologist Zuzana Jurková (2004, 2005). Hungary is also quite close culturally, and the Romani Pentecostal movement and songs were discussed there by several authors (Lange 2003b; Povedák 2015). Barbara Rose Lang has been interested in the influence of Christianity on the Romani in Hungary since 1992, and her interest later also encompassed gospel music, resulting in a monograph she wrote in 2003.

The consequences of the spreading of the Pentecostal movement in the communities of French Gitanos are discussed by Dienstbierová (2008). The main characteristics of Romani religious conversion and new piety, with special emphasis on Romania, are treated by László Fosztó in his publications (Fosztó 2009, 2019). The music of Pentecostal Romani in Portugal and Spain, later also in Angola, was investigated by Ruy Llera Blanes (2005, 2008). In her monograph, Tanya Riches provides an ethnographic account of three Australian Pentecostal congregations with Aboriginal senior leadership (Riches 2019).

By examining and comparing these sources, we will try to find an answer to the question of why the Romani are inclined towards the Pentecostal movement, what the soundscape of Pentecostal worship looks like, and what creates it.

## 2. Missionary Activities among the Romani in Slovakia

In the past, the Romani in Slovakia identified with the prevailing religion, mainly with the Roman Catholic Church. However, their active approach to the church community was minimal and often limited to basic rites of life, namely baptism, marriage, and funerals (Horváthová 1964; Davidová 1998; Kováč and Mann 2003; Podolinská and Hrustič 2010; Belišová and Mojžišová 2014). Their lukewarm attitude towards the Church might have been caused by a number of factors. On the one hand was the lack of interest of priests and religious communities in working with the Romani, in some cases even excluding the Romani and banning their entering the churches (Tatár et al. 2002, p. 17; Belišová and Mojžišová 2014, p. 91). Another reason might have been the different cultural disposition, the inappropriate style, and the process of the worship services.

The situation began to change after the restoration of religious freedom in 1990[2], when various church denominations, religious organizations, and non-profit organizations began to pursue missionary activities in Romani communities. Besides providing spiritual support and pastoral care to the Romani, the aim of these missions was to provide support and assistance to them in the social field, in the field of education, and in healthcare especially in locations where the Romani were marginalized or excluded. Along with an interest in the problems of the everyday lives of individuals, these missionary activities led to religious conversion, mainly to Pentecostalism, in many Romani communities. Evangelizing activities were first carried out among the Romani by the Apostolic Church, which is the only state-registered church of Pentecostal charismatic character in Slovakia besides Christian congregations. Since 2010, when there were only two separate Romani congregations in Slovakia, their number has grown to fourteen. They are all located in Eastern Slovakian regions—in the Košice and the Prešov Region.[3] In the second half of the 1990s, the international Pentecostal movement Devleskero Kher (Word of Life) began its missionary activities among the Romani, now focusing mainly on Slovak Romani congregations in England. In 2005, a Romani pastor, Emil Adam, came to Slovakia from Ostrava, the Czech Republic and, after months of mission in Romani slums, he founded a new congregation, Maranatha, with its center in Spišská Nová Ves, which later spread

to other places as well.[4] During the week, worship services are also held in individual slums, where the pastor or one of his assistants—deacons or future pastors—go (Podolinská and Hrustič 2010, pp. 27–28; Hrustičová 2010, p. 335; Hrustič 2014, pp. 198–99). Even in places where the major traditional churches—the Roman Catholic Church, the Greek Catholic Church, and the Evangelical Church of the Augsburg Confession—are active, the worship services where the majority of the participants are Romani are adapted to their temperament and include several practices that are characteristic of Pentecostal churches, including hymns—worship music. Several of the above denominations are also active in some Romani localities. However, it should be noted that, although the expansion of the Pentecostal movement in Slovakia and in other post-communist countries is tied to the period after the fall of communism, its beginnings reach back to much earlier, in some countries to the end of the Second World War (Slavkova 2012, p. 37).

The Pentecostal movement derives its name from the Jewish holiday of Pentecost, when the Holy Spirit descended on the first Christians, and they began to speak in unknown, mystical languages. Pentecostal church communities form a broad and diverse group of Christian churches and denominations, and their roots reach back to the Pentecostal movement[5] that places great emphasis on the experience of the Holy Spirit and the emotional internalization of this experience. When characterizing Romani Pentecostal communities in Slovakia, it is important to realize that they not only are influenced by Pentecostal practices, but also integrate Romani cultural elements, language, music, and traditions. The most important features of Pentecostal communities include the recognition of spiritual gifts with an emphasis on the gift of healing and the so-called speaking in tongues (glossolalia). Both of these manifestations are often practiced directly during the worship services. Along with the informality of the worship services and the personal freedom in faith, it is the emphasis on the personal spiritual experiences of the believers, their spiritual rebirth, prayer, spiritual and physical healing, and their missionary work and active praise of God that has probably attracted the Romani to this movement and stirred a great wave of conversions not only in Slovakia, but all over the world. Interestingly, in Western European countries,[6] the Pentecostal movement was spreading at the same time as in post-communist countries where religious freedom was restored[7].

### 3. The Formation of Pentecostal Romani Music

The conversion of Romani believers to the Pentecostal movement launched a long, creative process of the formation of Pentecostal Romani music. Music plays a vital and vibrant role in Pentecostal churches and is often considered a crucial element of worship and spiritual experiences. Pentecostal music is an important part of the evangelization process that goes beyond individual religious communities, regardless of whether the people are members of different denominations, and it delineates a group of believers through the dissemination of songs via the internet and the media (Slavkova 2012, p. 36).

Music plays an important role in all spiritual communities in the world. Already in ancient China and India, music was given a mystical meaning, and it plays an important role in meditation and prayer in Taoism, Buddhism, and Hinduism and in rituals and religious rites in Africa and among Native Americans. It is a part of the liturgy of all Christian denominations, and its significance is documented by Biblical texts, too. Most of the texts about praising God with music and singing can be found in the Psalms. The very word psalm comes from the Greek word psaltery, which was an old string instrument. Christians are also encouraged to sing psalms in the New Testament: "Let the word of Christ dwell in you richly in all wisdom; teaching and admonishing one another in psalms and hymns and spiritual songs, singing with grace in your hearts to the Lord." (King James Bible Online, Colossians 3:16). The texts of the Psalms themselves encourage people to praise the Lord with hymns "Praise the LORD with harp: sing unto him with the psaltery and an instrument of ten strings." (King James Bible Online, Psalm 33:2).

In Pentecostal churches, music enables the believers to express their emotions and their deep relationship with God. These are often passionate and emotional expressions of

faith through singing, playing musical instruments, and praise. Music serves as a means to lead prayers, praises, and worship during the worship services, but it is also a way to outwardly present the converts' belonging to the Pentecostal community. In addition to Sunday services, most Pentecostal communities organize regular gatherings two to three times a week in people's homes, with a lot of singing as well, and these are powerful ways to profess their faith before their neighbors and relatives. Over time, these songs have penetrated into the repertoire of everyday life, and the religious Romani sing them even outside their spiritual gatherings and prayers. They are sung by groups of young people even outdoors, and thus the songs become a kind of an identification mark of converted Romani. If the songs are musically attractive, they are also disseminated among other Romani.

> *We sing even at home. At home throughout the day, like that. All of us. My wife sings. When she starts, I join in. When I start, she joins in. Only about God. It's like prayer. Because there are people who understand it better through songs than if somebody explains it to them.* (Interview with Ľubo Mucha, Pastoral Centre of the Romani, Čičava, Vranov nad Topľou district, Greek Catholic Church 2012)

In some Pentecostal communities, music is considered to be a way to manifest charismatic gifts, such as speaking in tongues or prophesying. Musicians and singers are convinced that their musical talents are also a gift from the Holy Spirit, and many of them claim that they could not play or sing before.

> *That's how it happened then. The Lord really gave youth who simply began to play out of the blue. Then we began. Everyone received some song, some gift... Then he started... he also played the guitar, then also the drums, and then it just went on.* (Interview with Jozef Demjan, Baptist Union, Cinobaňa, Poltár district 2013)

> *We couldn't even play. But as soon as we believed in the Lord, we said we wanted to play about God. Now we have an organ, an electric guitar, and drums.* (Interview with Ladislav Feko, Brethren Church, Lesíček, Prešov district 2012)

Stereotypically, and even archetypally, the majority views the Romani as good musicians. Belonging to a musical family brought social and economic advantages to the Romani even in the past, as they could make their way to noble courts and to the wealthier strata of the population through their art. Most Pentecostal singers are naturally gifted and do not have a formal musical training. There are also some who had been professional musicians but gave up their profession to become musicians who play only hymns (Slavkova 2012, p. 43). However, when they convert to the Pentecostal movement, many musicians distance themselves from their musical past and divert attention from their own person, worthy of admiration as an artist, to God, as they feel that they themselves are only an instrument for God's praise.

> *I was a musician. I played music for sixteen years. And I thought I was popular, and that the world was praising me, like that, but I was zero, God didn't praise me. But now I know that God praises me that I praise Him, how I now play for Him, and I know that God praises me for that. That this glory is eternal, not like worldly glory, which only people praise.* (Interview with Stanislav Mižigar, Apostolic Church, Žehra, Spišská Nová Ves district 2012)

Several authors have pointed out the importance of Pentecostal praise in the lives of Romani believers and how the overall character of this religious movement was transformed through music (Slavkova 2012; Jurková 2004; Lange 2003b; Åberg 2014). Some are of the opinion that just as music is extremely important for the Romani in their everyday lives, the converted Romani have felt a need to bring music also into the worship services that have become an important part of their lives; they themselves have, therefore, shaped the character of the worship services. The services thus transformed, full of music that is close to the Romani, becoming attractive even for other believers, partly because of their music. In the first phases of the emerging congregations, the Romani sang the same

hymns during the worship services as the majority. They sang hymns from Slovak liturgical hymnals and so-called new Slovak hymns, which are popular mainly among young people and form part of informal Christian gatherings (Mináč 2003, pp. 41–42). Over the course of time, they began to adapt the music and the texts of the hymns. Changes of the texts to Romani language were sometimes deliberately initiated by non-Romani missionaries to bring the worship services closer to the Romani. They also published hymnals with a missionary purpose, which contained original evangelical hymns in the Slovak language, a few hymns translated into the Romani language, and a few that had Romani melodies and texts (Giľa nevo džDivipen 2000).

A more prominent change occurred when the Romani began to introduce their own melodies into the worship services, initially with very simple texts, which were sometimes only modifications of their traditional songs, e.g., instead of "my dear one is knocking on the window", they sang "Jesus is knocking on your heart", or they simply replaced the lover with Jesus in traditional love songs. Later, however, they began to compose their own songs, with original lyrics and music, and some communities even published their own liturgical songbooks for internal use, usually without notation because the melodies were well-known to everyone, as they were often songs composed in the given spiritual community (Spevník 2012).

> *"We adjust the praises we have to the time we live in. So not only classical, Romani, but we also make songs ourselves, and we also take songs from America, English songs, and we mix these."* (Interview with Rinaldo Oláh, Apostolic Church, Sabinov 2012)

> *"So we came up with some of the songs, we borrowed some from others who sing Romani songs, and we made them gospel-like."* (Interview with Martin Jano, Salesians, Košice Luník IX. 2012)

From the musical point of view, the Christian songs of the Romani are very heterogeneous. Many of them draw from the stylistic layers of their *neve giľa* (new songs)[8] and are inspired by various styles and genres of popular music just like these (Belišová and Mojžišová 2014, p. 49). This style, characteristic not only for Romani religious music, is sometimes referred to as *rom-pop* (Belišová and Mojžišová 2014; Belišová 2015; Mináč 2003; Slavkova 2012). Other sources of inspiration include some universal new hymns, but also traditional Romani melodies, whether dance-like ones, e.g., *čardaša* or, less frequently, *halgató*.[9] Simultaneously with the formation of Romani Pentecostal congregations, some Romani musicians emerged who are also pastors, e.g., Stanislav Mižigar of Žehra in the Apostolic Church, who was once a successful musician, whereas he is a pastor today. The process is sometimes vice versa, and a pastor composes new songs for the needs of the congregation and ultimately becomes successful in this area. One example is Emil Adam, the Romani pastor of the Maranatha community in Spišská Nová Ves. Even non-Romani pastors are often inclined towards Romani music, such as the Greek Catholic priest Martin Mekel, who not only composes Romani hymns, but even initiated the formation of several Romani gospel bands (F6, GPS) and launched the Romani gospel festival Festrom in 2006. Ricardo Kwiek is one of the most admired and most discussed authors and performers from abroad[10]. The combination of a pastor and musician in one person increases the respect of the believers.

## 4. Music as a Form of Prayer—Romani Praise Songs

Singing and music, and not only in the Romani community, are viewed as a form of prayer that can be very personal and profound, as music adds an emotional dimension to the words and helps create a connection with the spiritual world. Analogously to music, a religious community can be viewed as a symphony in which every part or musical instrument is irreplaceable. With their life, every person creates their own "melody", which can be attuned to God and enter into harmony with him. Even the sound of nature around us creates a perfect harmony, and not only a sonic one, which reflects God (Halpern and Lingerman 2005, p. 50).

*Singing is one of the most essential elements of worship. . . To open the gates of trust in God, nothing can replace the beauty of human voices united in song. This beauty can give us a glimpse of "heaven's joy on earth", as Eastern Christians put it. And an inner life begins to blossom within us.*[11]

A major part of Pentecostal Romani songs can be classified among so-called praise songs, sometimes not quite correctly referred to as gospel music. Gospel music is associated with the music of African American Baptist congregations in the USA. These songs, which have elements of folk music, ragtime, blues, and jazz, were sung by soloists or groups at various religious gatherings (Jakubíková 2006, p. 166). In the Slovak milieu, the first systematization and terminology of this genre was developed by the priest and journalist Juraj Drobný (1996, p. 21). The very name of praise songs or gospel music expresses the joyful content of these songs, which contradicts the image of God in traditional Romani songs. In the latter, God listens, and so the singer speaks to Him in the songs, but he is strict, retributive, and distant. Jesus does not appear in traditional songs at all (Belišová and Mojžišová 2014, pp. 22–28).

Marel o Del, marel,

kas kamel te marel.

Se man o Del mardža,

bo na šundžom la da.

[God punishes, punishes,

whomever He wants to punish.

God has punished me, too,

because I didn't listen to my mom.]

(recorded in Abranovce, Prešov district 1988).

In their new hymns, the Romani express their faith, joy, gratitude, and devotion to God, and they communicate with God. Just as in non-musical prayer, Romani praise songs also encompass various aspects of prayer, mainly the glorification and praise of God, petition, gratitude, and confession of sins.

Tu sal o Del, bararas tut,

Tu sal o Raj, ašaras tut,

Tu sal amaro Dad,

Savore avas angle Tute, amen

[You are God, we worship you,

You are God, we praise you,

You are our Father

We all come before You, amen.]

(Worship of God, recorded in Hlinné, Vranov nad Topľou district 2012)

Joj, Devla, joj, Devla,

Spomožin amenge.

Amenge, savore Romenge,

Pre oda šukar baro svetocis.

[Oh, God, oh, God,

Help us.

Us, all the Romani,

In this beautiful big world.]

(Petition, recorded in Rankovce, Košice district 2009)

Paľikerav tuke, Devla,

Vaš koda lačhipen, so tu kerdžal.

[Thank you, God,

For Your goodness, for what You have done.]

(Gratitude, recorded in Soľ, Vranov nad Topľou district 2012)

Devla, odmukh tu mange,

Mire vini na lačhipen,

The na lačho dživipen.

[God, forgive me,

My sins are bad,

I don't lead a good life.]

(Asking for forgiveness, recorded in Sabinov, Prešov district 2012)

One of the significant changes in the behavior of new Christians is their attitude towards so-called "secular music", i.e., to the music they liked, played, and sang before they became believers. After their conversion, they renounce their "secular way of life" and start listening to, admiring, composing, and playing only Christian music, as only that brings true joy to the soul (Slavkova 2012, p. 39). In their testimonies, Romani Christians emphasize that praise songs are not ordinary songs but a way of prayer (Kajanová 2009, p. 153) and glorification of God. The musical ensembles are then called "praise bands", and the performers "praisers".

*We played before, too. Mostly at discos. Now we play for God. It gives us joy and peace.* (Interview with Ľubo Mucha, Pastoral Centre for the Romani, Greek Catholic Church, Čičava 2012)

*It's not some worldly band, it's a praise band that serves God. Because the Bible tells us to praise the Lord, to worship the Lord, with guitars, trumpets, percussions, and so on. Everything that breathes. . . But it's used only for God's glory, not for worldly glory.* (Interview with Stanislav Mižigar, Apostolic Church, Žehra 2012)

*It's about man singing. Singing for the praise of God. It gives us joy. We are such a nation that when we sing, we sing from all our hearts. . . It's not that we're important. We play for the Lord. And that the Lord God is exalted.* (Interview with Ivan Čonka, Brethren Church, Vítkovce 2013)

The fact that most praise bands do not give themselves a name also arises from this humble attitude. To distinguish themselves, they are called according to the locality in which they live or perform, e.g., "the Praise Band of Žehra", praise bands from Sabinov, Žehňa, Bystran, and so on. Their names sometimes express a Biblical symbol, such as "Lačho lav" [Good Word], "F6" (according to Chapter 6 of the Epistle to the Ephesians from the Bible, which talks about the armor of God: truth, righteousness, readiness, joy, and faith), "GPS", and so on. The approach to the authorship of the songs and their use by other church communities reflects the philosophy that praisers play for God's glory, not their own. They often provide access to their albums free of charge on the internet, and one band's original songs are commonly played by other bands, too (Belišová and Mojžišová 2014, pp. 32–33). In their musical instruments and playing style, these bands resemble contemporary, so-called rom-pop bands. The most frequently encountered electronic musical instruments are keyboards and solo and bass guitars. Bands are usually supplemented by percussion and acoustic instruments such as the saxophone, the accordion, or bowed string instruments. Just like rom-pop bands, gospel bands are also made up of mostly male musicians, but the singers are often women.

## 5. The Soundscape of Religion

In Romani Pentecostal communities, we can observe the fusion of Romani cultural identity with general Pentecostal practices. The integration of Romani cultural elements,

language, and traditions into worship services can create a distinctive religious experience that reflects the unique identity of the community. There is no single uniform "sound" that would represent the religious practices of all of the Romani, as religious expressions may vary between different communities and individuals.

The relationship between sounds, emotions, the body, and space in Romani Pentecostal communities in Slovakia is a complex interplay that encompasses cultural, religious, and social dimensions. Pentecostalism is a form of Christianity that emphasizes a direct experience of the Holy Spirit, which is often expressed through ecstatic worship, speaking in tongues, and a lively and emotional style of religious expression. The sound of Romani Pentecostal worship is very varied and emotional. In addition to the music and singing, which can be heard almost throughout the worship service, the sounds associated with Romani spirituality often include glossolalia[12]. The sounds of individuals speaking in unknown languages are believed to be a manifestation of the Holy Spirit. This phenomenon contributes to a unique listening experience during the religious gatherings.

Pentecostal services also include other patterns of speech that affect the soundscape of the spiritual gatherings. Pentecostal communities remain united in their basic doctrinal ideas and one of their unifying features is the important role of a charismatic and authoritative pastor who has great respect among the community members (Slavkova 2012, p. 38). Missionary work among the Romani is not something that can be learned mechanically, and it certainly cannot be performed effectively just by being assigned to a Romani location. Experience shows that missionary Romani centers are established in places where pastors are internally motivated to work with the Romani and view this work as their mission. As a non-Romani pastor put it: "Let's not send anyone to the Romani by force because the meaning, effectiveness, and missionary dynamism of our presence will disappear [...] My life would perhaps be calmer and easier without the Romani, but it would be emptier, too." (Bešenyei 2009, p. 8). The pastor trains spiritual helpers from among the believers, who can eventually grow into pastors themselves. Worship services are usually led by one or more pastors who communicate with the faithful in a special form of "call and response". The pastor prays or interprets the Bible, and the faithful respond to him uniformly. These interactive elements can also be found in the liturgy of the major traditional churches, but in Pentecostal communities, they are more spontaneous and emotional and enhance the sense of community and oneness. Spontaneity, justified by "being led by the Spirit of God", inspires some believers to publicly share their experiences about God and to pray publicly. Prayers for healing create a very emotional soundscape. Towards the end of the service, several people are chosen to whom the faithful come to ask for prayers. The entire space is filled with muffled prayer, interrupted by the sighs and cries of people feeling healed or encouraged.

The vocal and physical manifestations of the faithful sometimes go into a trance-like state. When praying or giving testimonies, words lose their meanings, and the ritual itself becomes important. This varied mixture of sounds—permanent musical accompaniment as a continuous soundscape of the service, spontaneous vocal expressions of agreement with the pastor or the praying person, the ritual of praying for healing, expressing emotions through crying, sighing, laughing, or shouting—all of this leads to a gradual emotional and sonic unity of the community. This direction towards sonic harmony may be compared to improvisation in music when individual elements achieve harmony by listening to each other sensitively. This harmony does not come immediately but only after some time, along with singing, playing, and rituals.

The worship service held in the Romani part of the small village of Žehra in Eastern Slovakia, in which about forty people participated, had a similar atmosphere.[13] After converting to the Apostolic Church, the religious Romani converted a former pub into a place for their religious gatherings. The pub as a religious space had a symbolic meaning for them because many Romani overcame their addiction to alcohol after their conversion. Instrumental music filled the air during the services as if in waves. Romani musicians prefer to play in minor keys, and they usually play several musical instruments, but the

electric keyboards at this gathering maintained a uniform line even during the quieter parts when the music provided only a background to the pastor's sermon and prayer or to the testimonies and prayers of the believers. The speakers spoke with pauses, and the music grew louder during these. The music purposefully emotionally underlined the words of the prayer and, in turn, the intonation of the speech during the prayer followed the musical line to a certain extent. Musicians and speakers thus supported each other and united in emotion. They also worked with dynamics—as the music grew louder, so did the voice of the prayer. It was not possible to find out who was actually conducting this "dynamic conversation", whether the prayer (pastor) or the instrumentalist musician. Both music and prayer were taking place dynamically in waves of strengthening and weakening.

Into these dynamic waves of music and prayer, one male and one female voice sang a solo melody on meaningless syllables. The voices were quite distinct, and the accompanying music and the sung melodic line drowned out the sound of the prayer at some point, so that the words could not be understood at all. It did not seem to matter much, though, because the pastor cyclically alternated several topics, returned to them, and repeated words and sentences until they sounded almost like mantras. The pastor let himself be carried away by his emotions; he sometimes even shouted, and not because he wanted to drown out the other sounds. The pastor is considered a good singer and musician; he has a musical background in a secular band, which he left after his conversion, and now he utilizes his musical talent only in the church.

> *Christ's love, so that it bound me as well, so that you're saved, so that you have eternal life. So, Lord, we thank you, Lord, that we could meet here, that we can praise you, that we can exalt your name, that we can tell how wonderful you are, hallelujah, in your name… as your Spirit comes, as it magnifies, we also thank you, Lord, that we can invoke you…* (Interview with Stanislav Mižigar, the author's own fieldwork in Žehra, Spišská Nová Ves district, 2012)

The size of the gatherings varies from mass events in community centers attended by hundreds of people from the nearby areas, through smaller local gatherings with dozens of people, to meetings in homes where about twenty to thirty people can fit. Smaller gatherings provide more intimacy, and spiritual and emotional union arises more easily.

## 6. Emotions, the Body, and Movement

Music, prayer, emotions, and the body can all be closely connected, influence each other, and work together to create deep and meaningful experiences. Emotions promote a sense of spiritual connection and transformation through spiritual experience. Pentecostalism places a strong emphasis on emotional expressions in worship. When they are moved by emotion, believers often use their bodies through physical manifestations, such as shouting, crying, dancing, clapping, and laying hands during prayer. The physicality of worship is viewed as a way to bodily express the presence of the Holy Spirit and the believer's devotion to God. In a religious context, body language and gestures can have symbolic meanings. For example, raising hands in prayer or kneeling can represent submission to God.

The body can respond to religious emotions in different ways, while the hands play a special role. In many religions and spiritual traditions, the hands have a special meaning in the context of prayer. The way the hands are used can express devotion, humility, unity, and so on. In traditional Christian denominations, it is common to have the hands folded in front of the chest when praying, with the fingers sometimes intertwined. This attitude symbolizes devotion and humility, but also immersion into the self. The head is bowed, and the whole person is as if curled up and enclosed. In Pentecostal churches, however, the position of the hands is in contrast to the traditionally folded hands, as the believers often raise their hands above their heads with their palms open, as if directed towards the sky, and they complement their expression of openness by facing upwards. The position of the hands, the head, and the whole body is open. Sometimes they raise only one hand and put the other hand on their chest as if they wanted to deposit in their heart the supernatural

that they received from above, from God. The hands have a special role in healing rituals when the pastors place them on the heads of the believers while praying. In this case, the hands play the role of a mediator between the power of the Holy Spirit, who can help the believer recover from illness or resolve other problems in life through the hands of the praying person. During this ritual, believers no longer raise their hands high above their heads but place them at the level of their torso in front of them; their open arms symbolize their openness to receiving the power of the Holy Spirit through the praying person.

Another powerful means of expressing the emotions associated with prayer are facial expressions and the direction of the face. As mentioned above, facial expressions and the direction of the face complement hand movements and other bodily expressions. The most common facial expressions are smiling, which is associated with joy and gratitude, crying, which, however, does not express only sadness, or a meditative facial expression that points to the introspective moments of the believer or their thinking deeply about some ideas. The face can also reflect concentration and humility, but also ecstatic rapture often associated with other body movements.

People do not sit much during the worship services. Instead, they stand and sway to the rhythm for a major part of the service. A gentle swaying of the body can turn into dance movements. Dance is how the whole body responds to emotions. Romani Pentecostal communities commonly dance during their worship services. The musical performances and the singing of songs are accompanied by dancing and jumping to the rhythm of the songs, and this expresses the dancer's joy at the presence of the Holy Spirit and at the company of the other believers (Slavkova 2012, p. 41). Just like music and singing, spiritual dance is perceived differently than dance in a secular setting. However, at worship gatherings, it is usually not dances by couples but dances by individuals, by which they express their joy and sense of freedom. Pentecostal Romani view music and dance as a form of manifestation of the Holy Spirit and a way of unification among believers. Romani churches try to impose the idea that their followers devote themselves only to the worship of God, i.e., they only play and listen to religious music and have renounced secular music and traditional dances (Slavkova 2012, p. 40). However, this is not necessarily so in real life.

And here we arrive at the concept of purity and impurity, of pure and impure movements and pure and impure body parts. In addition to using the body to express piety or various emotions associated with spirituality, there is also another point of view on the body as a source of temptation and sin, referring to some Biblical verses[14]. Through faith and the action of the Holy Spirit, these manifestations can change to something positive, but this requires a certain discipline of the body, e.g., giving up alcohol, criminal activities, and violence (Roman 2017, p. 258). Several authors point out a change in the behavior of the believers after their conversion, by which they come closer to society's expectations (Podolínska 2014; Marushiakova and Popov 2014; Ripka 2015). Even the Romani sometimes considered themselves "better".

> *And that's what's essential, that man knows what is good and what is evil. There we didn't know what was good and what was evil. It was all the same to us even when we hurt someone. We were even worse. Now we know that we cannot do evil. And thank God that we accepted the Lord Jesus.* (Interview with Ivan Čonka, Brethren Church, Vítkovce 2013)

The perception of the body as a source of sin and temptation is in contrast to its perception as a means that facilitates a more intense experience of worship. This dilemma is partly explained by the concept of purity and impurity. In some Romani communities, e.g., in Finland (e.g., Viljanen 1974) or Hungary (Stewart 2005), the body is divided into a pure upper half and an impure lower half. Among the Hungarian Romani, the concept of purity is symbolically connected with shame for bodily reproduction in favor of a higher form of social reproduction. Such a perception of the body may explain their different relationship to corporeality (Stewart 2005, p. 185).

Therefore, "religious" body movements should be concentrated on the upper half of the body, especially on the hands, the head, and the face. Excessive movements of the hips

were not considered desirable in worship services, as they were associated with secular dancing. For example, when Finnish Romani believers dance, the "impure" area of the pelvis is not activated much (Roman 2017, p. 259). A similar perception of the body, applied mainly to women, can also be found in the Romani communities of other countries, but the strict rules have changed over time. For example, in Bulgaria, in the 1980s and 1990s, dances were banned in some Romani churches because they were considered secular. Clothing had to cover the whole body, and women had to wear white headscarves. Nowadays, conservative ways have changed and both singing and dancing are allowed (Slavkova 2012, p. 38). In Slovakia, dance movements are a common part of the worship services.

The premises where worship takes place also play a significant role in terms of acoustics. We can divide them into sacral and profane and into interior and exterior spaces. Sacral spaces do not have their own tradition in Romani worship services, as their sacral buildings were almost always newly built structures. The Salesian priest Peter Bessenyi who, with almost no state support, built churches on his own with the help of the believers in three places—Jarovnice, Bardejov, and Košice—stands out in this regard. The construction of a new prayer hall of the Apostolic Church in Sabinov is another significant achievement. Pastoral centers, schools, and nurseries are usually also built next to sacral buildings. Other congregations buy an old house and convert it into a prayer hall, or they rent sports halls or community centers. During the week, smaller worship services are held in the homes of believers. Outdoor services have a special significance. Sometimes, they are a necessity because the believers do not have a prayer room, and they are not allowed to use the village church, or the event is an outdoor one, like a pilgrimage, a concert, or a festival of religious music. Even in the case of profane or neutral premises, they become sanctified by the religious rituals themselves and by the way the believers perceive the space.

> *Because we always did such evangelizations outside, that is, directly in the slum. Or we did evangelization also in the community centre, we invited people there. And in various ways, like that. Actually, we still had very poor premises, there was an old house in this place, so we used it for sleeping when we did children's events at weekends. These premises were built by the Romani from among us, from our own community. Except for the roof, we did everything ourselves.* (Interview with the Greek Catholic priest Martin Mekel, Pastoral Centre for the Romani, Čičava, 2012)

## 7. Discussion

At the outset, we asked why the Romani are inclined towards the Pentecostal movement. I dare to argue with the opinion that it is music that attracts Romani believers to Pentecostal congregations. Although, in their present form, Pentecostal services create an attractive scope for Romani believers, mostly it is up to the Romani themselves, who first adapted some songs and later created songs of their own, what they look like, and what music is played and sung there.

The reason for their inclination to the Pentecostal sound may lie in the emotions it arouses in them and in the freedom of expression, both in terms of sound and movement. Therefore, even in communities of Romani believers who have remained in traditional denominations, e.g., in the Greek Catholic, the Roman Catholic, or the Lutheran Church, we can observe a similar course of the liturgical services with the infiltration of Pentecostal elements such as glossolalia, dance movements, raising and laying of the hands, and a repertoire of Romani songs.

Romani people often refer to their previous spiritual lives symbolically as dead. Many say that they did not understand anything before, that the services and sermons did not touch their lives, and that it was only after their conversion that their eyes and hearts opened. This may relate to the emotions that they did not feel before. In Pentecostal congregations, they can freely express their emotions, the believers have more personal contact with the pastors and with each other, who are interested in the problems of the individuals, pray for healing and for the resolution of their problems in life, and the ordinary Romani are actively involved in the worship services, prayers, and singing, which

strengthens their sense of oneness and responsibility. Faith penetrates their everyday lives also because, in addition to Sunday services, believers meet several times during the week in each other's homes. In the person of the pastor, Pentecostal congregations also substitute social work and counselling. When the Romani compare their old faith with their new one, they almost always talk about emotions—about joy and crying.

> *We cry for Jesus. We thank him for going to the cross for our sins. And we feel God is here with us. And when I used to go to church before, I sat, I listened, but I didn't know if I was filled with the Holy Spirit. And now I already feel that I am filled, I am filled with the Holy Spirit.* (Anonymous respondent during the worship service of the Apostolic Church in Žehra, 2013)

> *I was just touched by it, and I even burst into tears. Because there I could feel when he preached that Word of God, I could feel something special. And then I said to myself: "Well, this may be true. This may be the right way to God."* (Interview with Kveta Berkyová, Maranatha Church Community, Rudňany 2013)

We may establish that Pentecostalism is a dynamic religious force focused on music, songs and emotions.

> *This study has been written as part of the APVV-22-0389 "Research of Religiosity, Spirituality and Non-religiosity among the Roma in Slovakia" project.*

**Funding:** This research received no external funding.

**Institutional Review Board Statement:** Not applicable.

**Informed Consent Statement:** Informed consent was obtained from all subjects involved in the study.

**Data Availability Statement:** The data presented in this study are available upon request from the corresponding author because they are previously unpublished data from the author's research.

**Conflicts of Interest:** The author declares no conflicts of interest.

## Notes

[1]  Tolkien. https://en.wikiquote.org/wiki/The_Silmarillion (accessed on 25 February 2024).

[2]  In Slovakia, a so-called Velvet Revolution began on 17 November 1989, and it resulted in the fall of communism in Czechoslovakia. Communism had an atheistic ideology, and it significantly restricted religious freedom under the threat of various reprisals. After the fall of communism, the freedom to profess one's faith and practice religion without any interference by the state was restored.

[3]  Separate Romani congregations of the Apostolic Church in the Prešov Region are in Sabinov, Prešov—Archa, Soľ, Veľký Šariš, Humenné—Podskalka, Jarovnice, Kendice, and Ostrovany and in the Košice Region in Košice, Žehra, Bystrany, Michalovce, Pavlovce nad Uhom, and Slavošovce.

[4]  Currently, there are congregations of the Maranatha Christian mission in Spišská Nová Ves (this congregation gathers believers also from the neighboring villages of Markušovce, Vikartovce, Richnava, Smižany, Letanovce, Spišský Štvrtok, Hranovnica, Liptovská Teplička, Ľubica, Lomnica, and Vrbov, from where they are transported to the worship services in Spišská Nová Ves by buses), Poprad, Giraltovce, Humenné, Snina, and Liptovský Mikuláš, but even in Sheffield and Peterborough in England, where Romani have emigrated from Slovakia.

[5]  The origins of the Pentecostal movement date back to the nineteenth century, primarily to Germany and Great Britain, but it gained momentum in the twentieth century in Protestant Churches in the USA. https://www.britannica.com/topic/Pentecostalism (accessed on 12 November 2023).

[6]  E.g., in Hungary (Lange 2003a), Bulgaria (Slavkova 2012), the Czech Republic (Jurková 2004).

[7]  E.g., in Spain, Great Britain, Sweden (Roman 2017, p. 257), Finland (Åberg 2008).

[8]  In Slovakia, Romani songs can be divided into two major groups: 1. phurikane giľa (old songs) and 2. neve giľa (new songs) (Belišová 2012, p. 63). Neve giľa are inspired by various genres of popular music and are consequently also referred to as disco, sentimental ballads, tango, blues, funky, and so on. Literature also uses the term rom-pop (Belišová 2010).

[9]  Phurikane giľa (ancient songs) are divided into (1) halgató (protracted mournful songs) and (2) čardaša (dance songs) (Belišová 2012, p. 63).

[10]  Ricardo Kwiek, originally from Poland, is a popular Romani pastor and musician of international acclaim now living in Germany.

[11]  https://www.taize.fr/en_article338.html (accessed on 12 January 2024).

12    Glossolalia is the speaking of unintelligible, semantically meaningless syllables and sounds, speech without conscious intervention or intention. In charismatic communities and churches, glossolalia is considered to be a God-given ability. Metaphorically, it is referred to as the gift of tongues or speaking in tongues.

13    The author's own fieldwork in Žehra, Spišská Nová Ves district, 2012.

14    E.g., the Gospel of Matthew, Chapter 26 Verse 41 says: "Watch and pray, that ye enter not into temptation; the spirit indeed is willing, but the flesh is weak." (King James Bible Online 1970).

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
