# Peer review of "Sounds, Emotions, and the Body in Pentecostal Romani Communities in Slovakia"

_religions, doi:10.3390/rel15050532_

Round 1

Reviewer 1 Report

Comments and Suggestions for Authors

The title of the article is interesting, but it promise too much. It needs some background about the Pentecostalism and the the stereotypes about the Roma and their religious, even the material is empiric and basically good one

Referense are absolutely too old, there are many new research about the topic (not only Jurkova, Lang ect...)

The norms of the purity and inpurity is a little bit old too (Doglas, Viljanen ect...). We can call these as a Primordialist theory or functionalist...

There is a lack of the political movements, because religious among the Roma was very important what comes to the assimilation politics

The categorization to the neuvo Gila and staraja Gila is okay (Belisova, Kovalschik, Jurkova ect..) but the problem is that in a semantic level there are many references to the religious also in traditional and popular music, even Roma hip hop.

What comes to the empirical data - which is very claryfing - it would be pleasure to read more, how the Roma use different musical and unmusical elements construct their ethnicity (for example language). There are few, but I hope more.

There are many articles about the romani stereotypes, the writes should read some of them. It opens more widely the case, I hope

It should be noticed that the Pentecostalism opens more choises to Roma as well as other minorities, because it is not so dogmatic but based on personal experience, and that is why in many countires the Roma are on the stage..

Anyway, most of the texts in article was very interesting to read

Comments on the Quality of English Language

For me okay, but should check

Author Response

Review 1

Summary

Thank you very much for taking the time to review this manuscript and for your thought-provoking notes and suggestions. I have tried to incorporate them, at least partially, into my study. At the recommendation of you and other reviewers, I added or changed a few new paragraphs to the article, mainly on lines 38 – 52, 38- 49, 56 – 62, 64 – 69, 90 – 98, 138 – 151, 110 – 116, 138 – 151, 248 – 253, 402 – 408, 493 – 496, 652 - 653

Questions for General Evaluation

  1. Is the content succinctly described and contextualized with respect to previous and present theoretical background and empirical research (if applicable) on the topic?
  2. Are all the cited references relevant to the research?
  3. Is the article adequately referenced?

Reviewer’s Evaluation

Must be improved

Response and Revisions

I believe that all three comments of the reviewer are aimed at supplementing and expanding information about theoretical works and researches on the topic in question. At the beginning of the article, I added several authors whose research is relevant to my topic in the overview section.

Comment 1

Referense are absolutely too old, there are many new research about the topic (not only Jurkova, Lang ect...)

Response 1

In the text, there are also references to recent works, and I have now added some more of these to the introduction. The works I originally mentioned in the introduction appeared to be the closest ones to my chosen topic and region, although they are older ones.

Comment 2

The norms of the purity and inpurity is a little bit old too (Doglas, Viljanen ect...). We can call these as a Primordialist theory or functionalist...

Response 2

Thank you for the information.

Comment 3

There is a lack of the political movements, because religious among the Roma was very important what comes to the assimilation politics

Response 3

These contexts are noteworthy and important ones, but there is no room in this paper for a more detailed treatment of the development of religion among the Romani, nor was this its purpose.

Comments

The categorization to the neuvo Gila and staraja Gila is okay (Belisova, Kovalschik, Jurkova ect..) but the problem is that in a semantic level there are many references to the religious also in traditional and popular music, even Roma hip hop.

What comes to the empirical data - which is very claryfing - it would be pleasure to read more, how the Roma use different musical and unmusical elements construct their ethnicity (for example language). There are few, but I hope more.

There are many articles about the romani stereotypes, the writes should read some of them. It opens more widely the case, I hope

It should be noticed that the Pentecostalism opens more choises to Roma as well as other minorities, because it is not so dogmatic but based on personal experience, and that is why in many countires the Roma are on the stage..

Response

All your suggestions are very interesting and thought-provoking, and I will certainly utilize them in my subsequent treatment of this topic.

Reviewer 2 Report

Comments and Suggestions for Authors

This is an excellent description of Sounds, Emotions, and Body among Slovakian Romani. I agree with all your main points, with some reservations.

- -            “In Pentecostal churches, music enables the believers to express their emotions.” Absolutely. But also in non-Pentecostal churches.

-            “A more prominent change occurred when the Romani began to introduce their own melodies.” This is true in all dynamic churches.

-            “Music as a Form of Prayer.” Totally agree. But do the Romani come to church for the music, and then find it a form of prayer? Or do they come to church for prayer and also like music? Your conclusion suggests the first.

-            “The Romani songs can be classified as praise songs.” The “Praise and Worship” songs are popular in many churches, not just the Pentecostal ones.

-            “After their conversion, they renounce their “secular way of life.” This is true about conversions in all churches.

-            “The relationship between sounds, emotions, the body, and space in Romani worship is complex. But it is so in all churches, even those without glossolalia.

-            Your description of Emotions, the Body, and Movement is typical of all or most Pentecostal churches, but not unique to the Romani. In Kinshasa in the Congo the Mass lasts two and a half hours with high emotions, body movements, loud music and singing, and dance, and they are not Pentecostal.

 Your discussion unveils your implicit thesis which you already announced it in the Introduction: “In this paper, I ponder why the Romani prefer the Pentecostal character to the traditional way of worship.” In the conclusion, you state, “I dare to argue   with the opinion that it is music that attracts Romani” This is merely an opinion. It is based on the “post hoc ergo propter hoc” fallacy: after joining a Pentecostal church, they liked “the music as prayer,” therefore, you say that it is the music that attracted Romani.”

        What can you do? You can show that the Romani worship services are the same or different from other countries, especially the US and the Third World. This would be difficult.

        Or you can show what the Romani gained by joining a Pentecostal church. You have a lot of data from your research. You only have to make changes in the introduction and present all your points as spiritual benefits: in the past the Romani were passive attenders, but now they are engaged in their church. Now they find that music is important in worship, that music can be prayer; they sing prayers even at home; they can create their own melodies drawing on their subculture; after conversion their give up their former way of life; music is related to emotions and body movements. Conclusion: Pentecostalism is a dynamic religious force centered on music, songs, and emotions. Here you can also mention that Pentecostalism in the broad sense in a major force in the Global South, especially in Latin America.

As it stands, your paper is mainly a description of Pentecostal worship as in many parts of the World; the Romani are not very different. If changed, it would be something like “The Benefits in Sounds, Emotions, and the Body of Romani of Pentecostal Communities in Slovakia.” That would be special to the Romani.

Author Response

Summary

Thank you very much for taking the time to review this manuscript and for your thought-provoking notes and suggestions. I have tried to incorporate them, at least partially, into my study. I made small changes in the introduction and the conclusion. At the recommendation of other reviewers, I added or changed a few new paragraphs to the article, mainly on lines 38 – 52, 38- 49, 56 – 62, 64 – 69, 90 – 98, 138 – 151, 110 – 116, 138 – 151, 248 – 253, 402 – 408, 493 – 496, 652 - 653

Comment

Or you can show what the Romani gained by joining a Pentecostal church. You have a lot of data from your research. You only have to make changes in the introduction and present all your points as spiritual benefits: in the past the Romani were passive attenders, but now they are engaged in their church. Now they find that music is important in worship, that music can be prayer; they sing prayers even at home; they can create their own melodies drawing on their subculture; after conversion their give up their former way of life; music is related to emotions and body movements. Conclusion: Pentecostalism is a dynamic religious force centered on music, songs, and emotions. Here you can also mention that Pentecostalism in the broad sense in a major force in the Global South, especially in Latin America.

Response

Yes, I agree that the Pentecostal worship services of various ethnicities in various countries have many common elements. And that even non-Pentecostal services somewhere in Africa may have a similar character. However, the scope of my paper is not as broad as to include this, too. It focuses on Romani Pentecostal congregations in Slovakia. Based on my own experience and the interviews I conducted, I can confirm that the difference in Slovakia lies in intensity. The manifestations mentioned are a lot stronger in Romani congregations, and this is confirmed even by non-Romani believers. “You should hear how beautifully the Romani sing and how emotionally they pray.” They admire the Romani for being convincing and deeply emotional, which they attribute to their profound faith, but also, in the spirit of stereotypes, to their greater emotionality and musicality.

Since the paper focuses on the music and soundscape of Romani worship, I think a significant difference (besides the intensity, which cannot be quantified) is, for example, the penetration of Romani praises into everyday repertoire. They sing praises at home, just as they used to sing their traditional songs, which are almost completely disappearing from the repertoire of the converted believers. Another significant difference is the introduction of their cultural elements, I mean the melodies of their songs and the poetics borrowed from their traditional songs, and yes, even their Romani language. This could be an interesting impetus for further study and research.

Comment

Your discussion unveils your implicit thesis which you already announced it in the Introduction: “In this paper, I ponder why the Romani prefer the Pentecostal character to the traditional way of worship.” In the conclusion, you state, “I dare to argue   with the opinion that it is music that attracts Romani” This is merely an opinion. It is based on the “post hoc ergo propter hoc” fallacy: after joining a Pentecostal church, they liked “the music as prayer,” therefore, you say that it is the music that attracted Romani.”

Response

I might not have expressed myself clearly enough. I do not think music is what a priori attracts the Romani to Pentecostal worship. It depends on the situation and the music itself. It is definitely a set of several things. But one thing is certain from my observations and interviews: once they join these religious communities, the Romani transform their musical aspect significantly. And in this transformed form, they become attractive even to other Romani because of their music. Many, for example, recall how Pastor Emil Adam began evangelizing. He came from Bohemia to the poorest region of Slovakia, to Spiš, and organised concerts of Christian praises (already with Romani songs) in meadows near Romani slums.

Reviewer 3 Report

Comments and Suggestions for Authors

Thanks for this excellent and important contribution to Pentecostalism and music. Thoughtful discussion and analysis!

Consider the following:

In your introduction, more clearly define your own overall methodology. This additional information will help readers understand your field of inquiry and approach to these important questions. Also, consider adding relevant historical data on the Pentecostal movement in this region. Discuss questions such as where did it begin among the Romani and what are the roots and early influences. You linked the movement to the biblical origins (i.e., Pentecost), but what about the global and regional origins? 

Consider incorporating recent scholarship on Pentecostalism and worship music to distinguish the Pentecostal Romani Communities in Slovakia. Scholars such as Tanya Riches, Lester Ruth, Lim Swee Hong, Steve Felix-Yager, and Allen Anderson will help set your ethnographic work within the larger discussions on music within Pentecostalism. These could be drawn in your intro, analysis, or conclusions. 

Please share the size of the gatherings you describe around line 385. Discussion on the leadership and make-up of the musical ensembles would also enhance clarity and contextual understanding for your readers. 

In your conclusion, consider discussing transformation within "secular" spaces, which is mentioned throughout your paper.

Overall, thank you for a thoughtful, original paper.

Comments on the Quality of English Language

The use of incomplete questions in the introduction and conclusion may fit better for a presentation than a published article. Consider rephrasing these questions. 

Author Response

Summary

Thank you for taking time to read my paper and for all your suggestions. I have incorporated some of them into my study and added more detailed information. At the recommendation of you and other reviewers, I added or changed a few new paragraphs to the article, mainly on lines 38 – 52, 38- 49, 56 – 62, 64 – 69, 90 – 98, 138 – 151, 110 – 116, 138 – 151, 248 – 253, 402 – 408, 493 – 496, 652 - 653

Comment

In your introduction, more clearly define your own overall methodology. This additional information will help readers understand your field of inquiry and approach to these important questions. Also, consider adding relevant historical data on the Pentecostal movement in this region. Discuss questions such as where did it begin among the Romani and what are the roots and early influences. You linked the movement to the biblical origins (i.e., Pentecost), but what about the global and regional origins?

Response

I have expanded the information in the introduction about lines 53 - 68 and about lines 138 as you recommended.

Comment

Consider incorporating recent scholarship on Pentecostalism and worship music to distinguish the Pentecostal Romani Communities in Slovakia.

Response

Could you please explain what scholarship (research) you mean?

Comment

Scholars such as Tanya Riches, Lester Ruth, Lim Swee Hong, Steve Felix-Yager, and Allen Anderson will help set your ethnographic work within the larger discussions on music within Pentecostalism. These could be drawn in your intro, analysis, or conclusions.

Response

I have added a few more scholars about lines 70 - 116

Comment

Please share the size of the gatherings you describe around line 385. Discussion on the leadership and make-up of the musical ensembles would also enhance clarity and contextual understanding for your readers.

Response

I think you are asking about the size of the gathering in Žehra, which I describe and analyse in more detail. I have added this information to the text.

I have also added a few sentences about the composition of the musical ensembles.

Comment

The use of incomplete questions in the introduction and conclusion may fit better for a presentation than a published article. Consider rephrasing these questions.

Response

The questions are stated in the paper twice (at the beginning and at the end) by mistake. I have deleted them from the end of the paper and reformulated them at its beginning.